# Role of Non-Coding Regulatory Elements in the Control of GR-Dependent Gene Expression

**DOI:** 10.3390/ijms22084258

**Published:** 2021-04-20

**Authors:** Malgorzata Borczyk, Mateusz Zieba, Michał Korostyński, Marcin Piechota

**Affiliations:** Laboratory of Pharmacogenomics, Department of Molecular Neuropharmacology, Maj Institute of Pharmacology, Polish Academy of Sciences, 31-343 Krakow, Poland; michkor@if-pan.krakow.pl (M.K.); marpiech@if-pan.krakow.pl (M.P.)

**Keywords:** glucocorticoid receptor, GR, NR3C1, enhancer sequences, expression regulation, EP300, histone modifications, ENCODE, RNA-Seq, ChIP-Seq

## Abstract

The glucocorticoid receptor (GR, also known as NR3C1) coordinates molecular responses to stress. It is a potent transcription activator and repressor that influences hundreds of genes. Enhancers are non-coding DNA regions outside of the core promoters that increase transcriptional activity via long-distance interactions. Active GR binds to pre-existing enhancer sites and recruits further factors, including EP300, a known transcriptional coactivator. However, it is not known how the timing of GR-binding-induced enhancer remodeling relates to transcriptional changes. Here we analyze data from the ENCODE project that provides ChIP-Seq and RNA-Seq data at distinct time points after dexamethasone exposure of human A549 epithelial-like cell line. This study aimed to investigate the temporal interplay between GR binding, enhancer remodeling, and gene expression. By investigating a single distal GR-binding site for each differentially upregulated gene, we show that transcriptional changes follow GR binding, and that the largest enhancer remodeling coincides in time with the highest gene expression changes. A detailed analysis of the time course showed that for upregulated genes, enhancer activation persists after gene expression changes settle. Moreover, genes with the largest change in EP300 binding showed the highest expression dynamics before the peak of EP300 recruitment. Overall, our results show that enhancer remodeling may not directly be driving gene expression dynamics but rather be a consequence of expression activation.

## 1. Introduction

Glucocorticoid receptor (GR), encoded by the *NR3C1* gene, is a ligand-dependent transcriptional regulator. After binding to glucocorticoids, it translocates to the nucleus and interacts with the genome to influence gene expression. GR stimulation activates and represses hundreds of genes [1]. GR signaling plays an important role in the regulation of neuronal and glial cells, including their responses to drugs [2,3]. ChiP-Seq experiments allow for high-throughput analysis of chromatin binding, and studies with this method have brought great progress in understanding GR binding to DNA. The specificity of GR binding depends on pre-existing chromatin landscape, with factors, such as accessibility and motifs promoting the binding [4,5,6]. GR frequently binds to enhancers-*cis*-regulatory elements distal to transcription start site (TSS) of both repressed and activated genes [6,7].

Although it is difficult to define the precise features of enhancers, they are often associated with specific histone modifications and the presence of particular proteins [8]. Histone methylations and acetylations mark active and dormant enhancers. Monomethylation of lysine 4 and acetylation of lysine 27 of histone H3 (H3K4me1 and H3K27ac) mark active enhancers, whereas enhancers lacking H3K27ac are poised [9]. Another marker of active enhancers is the Histone acetyltransferase EP300 (also known as p300) that functions as a transcription coactivator [9,10]. Apart from enhancers, moderate EP300 enrichment is found in core promoters [11]. H3K4me1, on the other hand, is found predominantly in distal elements [9]. Enhancer sequences are also highly conserved [12,13]. Once an enhancer is active, it works to activate transcription of its targets via chromatin loops. Millions of such loops exist in the human genome [14]. The important physiological role of enhancers was confirmed by genomic studies that found many disease-associated polymorphisms in enhancer sequences [15,16].

GR effects on gene expression have been thoroughly investigated in several studies and extensively reviewed [17,18,19,20]. However, it is not known how enhancer remodeling induced by NR3C1 connects to gene expression dynamics. The interplay between NR3C1 binding, enhancer remodeling, and gene expression changes remain not thoroughly investigated, particularly at time points in the range of hours. Right after glucocorticoid (GC) treatment, NR3C1 binds to chromatin, and maximal binding in vitro can be detected after 30 min of GC exposure [6]. GR binds predominantly to distal sites that are typically already predetermined and marked as active enhancers [4,6,7]. Binding is followed by further enhancer remodeling by redistribution of EP300. In a recent study, McDowell et al. showed that changes in transcription factor (TF) binding and histone modifications within enhancers that occur after dexamethasone (synthetic GC) treatment correlates with gene expression changes [6]. They also showed that NR3C1 binding and EP300 mutually reinforce each other. Still, an outstanding question remains: how does NR3C1 binding to enhancers, EP300 recruitment, and histone modifications relate to the timing of transcriptional changes?

Here we present results of an investigation of the interplay between gene expression, GR binding, and remodeling of the enhancer landscape. The dataset used for this research was obtained from the Reddy Laboratory through the ENCODE project and includes both RNA-Seq and ChIP-Seq data of dexamethasone-treated human epithelial-like cell line collected at different time points between 30 min and 12 h [6,21]. The analysis started with the identification of genes robustly regulated by NR3C1. For each gene, dynamics of a single NR3C1-binding site outside of the promoter regions were analyzed. Apart from NR3C1 binding, we focused on markers of enhancer remodeling: EP300, H3K4me1, and H3K27ac. Time-course analysis of upregulated genes showed that in selected enhancers majority of NR3C1 binding occurs within the first hours of GC treatment, which is then followed by an increase in H3K27ac, a marker of active enhancers. Maximum expression dynamics occur ~1 h after the maximal NR3C1 binding is reached and nearly coincide with the largest EP300 recruitment. Additionally, genes with the largest change in EP300 binding were investigated and showed the highest expression dynamics before peak EP300 recruitment. Overall, our results show that enhancer remodeling may not directly be driving gene expression dynamics but rather be a consequence of or co-occur with expression activation.

## 2. Results

### 2.1. GR Regulates Expression Bidirectionally

Gene expression changes were first evaluated to indicate differentially regulated transcripts. As GR regulates hundreds of genes, the goal of this part of the analysis was to select a smaller subset of genes with clear and robust expression changes after dexamethasone [6]. After applying a detection threshold of FDR < 0.0000001, we identified 374 upregulated and 371 downregulated genes (Figure 1 and Appendix A). An extra cluster of random genes with similar gene expression levels and transcript lengths was used for comparison (Appendix A). Enrichr was used to identify published gene-sets with overlapping genes [22]. The list of significantly overlapping groups of genes includes 346 published gene-sets (adjusted *p*-value < 0.05). In the top part of the list, groups of steroid-activated genes, as well as transcripts regulated in hormone-dependent cancers, were identified [23,24,25,26]. What is more, similar to others, we observe that GR activation leads to bidirectional gene expression changes and that the number of up- and downregulated genes are roughly equal [1,6].

### 2.2. Gene Activation Is Associated with Enhancer Remodeling

We investigated the relationship between the observed gene expression changes and enhancers’ activity in the vicinity of those genes. Enhancer peaks were defined as ChIP-Seq peaks occurring within −/+ 100 kbp (kilo base-pair) from the TSS with the exclusion of the promoter region (−/+ 2 kb surrounding the TSS). If a gene had more than one peak, the one with the strongest NR3C1 signal was chosen. Two hundred thirty-six peaks of upregulated genes were identified at 1 h of dexamethasone treatment. Thus, 63.8% of the upregulated genes had NR3C1 signals. One hundred one peaks were identified near downregulated genes (in 26.9% of genes), and 183 out of 743 random genes had NR3C1 peaks (24%). The distribution of identified enhancer peaks showed that in the case of upregulated genes, the highest peaks tend to occur closer to the promoter region than for the other gene clusters (Appendix A). Still, these detected peaks are well outside the core promoters, defined as a region of a few hundred base pairs surrounding the TSS [27].

Next, we identified signals for EP300, H3K4me1, and H3K27ac in the same range as discovered peaks. Figure 2 shows the time course of averaged ChIP-Seq signal in identified peaks. For NR3C1, EP300, and H3K27ac, the signal was strongest in peaks near upregulated genes throughout the time course (two-way ANOVA, time:regulation, *p* < 0.02 for all four regulatory factors, Appendix A). EP300 binding near upregulated showed a notable decrease in signal at the 4 h timepoint. This could indicate overlapping of distinct patterns of EP300 binding, but individual time-courses did not confirm this observation (Appendix A). As downregulated genes did not show robust changes in the ChIP-Seq signal for any of the analyzed regulatory factors, they were not included in further analyses (Figure 2 and Appendix A). For all analyzed regulatory factors, similar dynamics were observed at promoter regions (Appendix A).

### 2.3. The Largest EP300 Recruitment Coincides with the Most Dynamic Gene Expression Changes

We interrogated the timing of gene expression dynamics concerning enhancer remodeling for upregulated genes. For gene expression, the time point of the largest dynamics was calculated as a weighted time average, where the first derivative of normalized transcript abundance was used as weights. For each of the analyzed ChIP-Seq tracks, maximum binding was interrogated by calculating a weighted time point with amplitudes at each time used as weights. The median weighted time point of maximum gene expression changes was 5 h and 27 min. Although increased and decreased gene expression is visible after 30 min of dexamethasone treatment, the changes do not reach their maximum dynamics for hours. The median weighted time point of maximum NR3C1 binding was 4 h 36 min. For H3K27ac and H3K4me1, the time points were 4 h 59 min and 4 h 48 min, respectively. Thus, gene expression changes reach maximum dynamics 51 min after maximum NR3C1 recruitment, and H3K27ac and H3K4me1 maximum amplitudes are observed within this interval. Most strikingly, the median of maximum EP300 binding nearly coincided in time with maximum changes in gene expression and occurred at 5 h 33 min. Thus, maximum expression dynamics occur together with the highest EP300 recruitment to peaks occupied by NR3C1 and are preceded with the highest NR3C1 binding and histone modifications (Figure 3).

### 2.4. Peaks with the Highest Changes in EP300 Recruitment Occur near Upregulated Genes and EP300 Is Recruited to Them after the Largest Expression Dynamics Are Reached

To further analyze the relationship between dexamethasone treatment-induced enhancer activity and gene expression, genes with the biggest change in EP300 binding in NR3C1 peaks were identified. First, a list of all identified NR3C1 peaks in enhancer regions was prepared. Then top 100 peaks with the largest difference in EP300 binding between any two time points starting from 30 min were selected. Then, for each of those peaks, the nearest gene was found. In the case two peaks were attributed to the same gene, the peak with the higher amplitude was retained. Expression dynamics of these genes were then analyzed, and 46 out of 100 genes were differentially expressed (FDR < 0.1), and 23 genes passed the previously applied 0.0000001 FDR threshold. Here, again genes were divided into up- and downregulated clusters (15 downregulated and 85 upregulated, Appendix A). We investigated the time points of the highest dynamics of enhancer remodeling of upregulated genes with the highest EP300-binding differences and compared them with upregulated genes from the gene expression-based analysis. Upregulated genes that showed the largest differences in EP300 binding in their strongest NR3C1 peak showed the largest expression dynamics 4 h 54 min and the highest EP300 binding at 5 h 38 min (Figure 4 and Appendix A, Appendix A). Thus, for the upregulated genes with the highest EP300 dynamics, the time that elapses between maximum expression changes and enhancer remodeling is even more uncoupled than for upregulated genes chosen based on their expression changes.

## 3. Discussion

Here we present results of an investigation of the time course of gene expression and chromatin-binding changes after dexamethasone treatment in a human epithelial-like cell line [21]. In the first part of the analysis, 374 genes upregulated after dexamethasone treatment were identified, while at the same FDR threshold, 371 genes were downregulated. These results are consistent with the original analysis of gene expression in the same dataset, although here, a more stringent statistical threshold was applied; thus, the number of genes in each cluster is lower than in the previous analysis [6]. For each gene, we identified a single ChIP-seq peak in the enhancer region based on the strongest NR3C1 amplitude. Each of the identified peaks was analyzed in terms of EP300 and NR3C1 binding and two histone modifications H3K27ac and H3K4me1. Next, enhancer-sited dynamics were interrogated in relation to gene expression dynamics and the time course of alterations.

Enhancer remodeling was analyzed for genes upregulated after dexamethasone as downregulated genes did not exhibit robust changes, neither in NR3C1 nor EP300 binding when compared to a randomly selected list of genes. This observation is in concert with literature [28]. The analysis of temporal order of changes in transcription factor binding and histone modifications revealed that there is no point in which all analyzed regulatory factors are orchestrated and acting as one machinery. It is rather a chain of subsequent events that could act through multiple mechanisms. Although strongly supported by the data, the above conclusions have limitations as they describe average dynamics of gene regulation and not individual time courses. We suggest that the binding of NR3C1 to its response elements activates at least two separate pathways. First, direct activation of expression, and the second, starting an orchestrated sequence of chromatin-associated events. The strongest binding of NR3C1 occurs just after dexamethasone treatment, and at this moment, transcription of target genes increases. However, at the following time points, binding of NR3C1 decreases, but the dynamics of changes in expression increase. The latter is associated with an increase in EP300 binding and H3K27 acetylation.

The association of elevated expression and increased binding of EP300 to enhancers made it the perfect marker of active enhancers, and EP300 is considered to be a coactivator of transcription [9,10]. As a result, most studies concentrate on the requirement of EP300 in the process of direct transcriptional regulation [6]. However, the results of this study suggest that active transcription is followed by EP300 recruitment. This observation adds another layer of meaning to the tight association of EP300 and enhancer-driven expression and shows that recruitment of EP300 might not be as crucial as previously thought. The enhancer-mediated transcriptional activity requires EP300-dependent H3K27 acetylation [29]. However, it might also be acetylated through other redundant mechanisms [29,30]. This may explain why mutations in EP300 lead to milder forms of Rubinstein–Taybi syndrome than mutations in CREBBP [31,32]. This discovery also points to the potential for a positive feedback loop. Activated enhancers may recruit EP300 and, in consequence, become more sensitive to further activations. More EP300 would promote further binding of TFs, including GR. Positive feedback loops were previously suggested as one of the cell differentiation mechanisms [33].

Close temporal association between EP300 binding and acetylation of histone H3 was previously shown [34]. However, for genes downregulated by dexamethasone, this association was not observed. While EP300 binding increases mildly even for downregulated genes, acetylation of H3K27 sites decreases. This suggests that for downregulated genes, there is a robust mechanism driving deacetylation that overcomes EP300-dependent acetylation. H3K27 acetylation is also, out of the four analyzed alterations, the one that directly precedes dynamics of changes in gene expression.

Primed enhancers are marked by H3K4me1 [35]. They are further activated by acetylation of H3K27 through, among others, EP300 acetyltransferase. However, the inverse mechanism was also postulated through which transcription factor recruitment can result in H3K4me1 priming [36,37]. Here, we observe a late increase in H3K4me1 after activation of NR3C1. This immediately suggests a possible mechanism of steroid-dependent cell differentiation in which repeatedly activated NR3C1 responsive sites would be primed by modifying the pattern of accessible enhancers for a particular cell type. This is in concert with the view of H3K4me1 as a fine-tuner of enhancer activity [38,39,40].

## 4. Methods

All of the code used to download, preprocess and analyze data are available in the project’s GitHub repository: https://github.com/ippas/ifpan-chipseq-timecourse (accessed on 10 March 2021).

### 4.1. Datasets

RNA-Seq (.tsv files) and ChIP-Seq (.bed and .bigwig files) data were downloaded from GEO database (https://www.ncbi.nlm.nih.gov/geo/query/acc.cgi?acc=GSE91222 (accessed on 10 March 2021) and https://www.ncbi.nlm.nih.gov/geo/query/acc.cgi?acc=GSE113464 (accessed on 10 March 2021); BioProject link: https://www.ncbi.nlm.nih.gov/sra?linkname=bioproject_sra_all&from_uid=356880 (accessed on 10 March 2021)) [21]. For ChIP-Seq, the following tracks were used in the analyses: NR3C1 (GR), EP300, H3K4me1, and H3K27ac.

### 4.2. RNA-Seq Analysis

RNA-Seq analysis was performed in R 3.4. First, quantile normalization was performed on raw counts. After normalization, one-way ANOVA with FDR correction was used to evaluate gene expression changes. Genes with FDR < 0.0000001 were included into the cluster analysis (*n* = 745 genes). Genes were classified as either up- or downregulated by dexamethasone. Additional random (control) gene-set of genes with similar gene-expression levels to regulated genes was selected. For dynamic expression analyses (Figure 3 and Figure 4), outliers in terms of gene expression counts were removed according to the boxplot.stats function in R. Weighted time point of maximum gene expression changes was calculated as a count-change-weighted average of time.

### 4.3. ChIP-Seq analysis

For each TSS, defined as the gene start in the Ensembl database, lists of surrounding (+/− 100,000 bp) peaks for NR3C1 were selected from *.bed files. Peaks common for samples within each experimental group were obtained using bedtools intersect. The peaks were further divided into core promoter regions (within +/− 2000 bp from TSS) and enhancer regions (excluding +/− 2000 bp from TSS). Within the enhancer region, a peak with the highest amplitude for each gene in 60 min time point after dexamethasone treatment was considered for further analysis. This time point showed the highest overall GR binding signal. For each analyzed NR3C1 peak, ChIP-Seq coverage for EP300, H3K4me1 and H3K27ac were analyzed within the coordinates of each NR3C1 peak using .bigwig data taking maximum coverage as summary value. For EP300 dynamics association with expression, a list of all identified NR3C1 peaks (60 min time point) in enhancer regions was prepared. Next, coverage of EP300 for those peaks was obtained using bigWigAverageOverBed from the kentUtils package. The top 100 peaks with the largest difference in EP300 binding between any two time points starting from 30 min were selected. Then, for each of those peaks, the nearest gene was identified. In case two peaks were attributed to the same gene, the peak with the higher amplitude was retained. For the temporal analysis, the time point of maximum binding was defined as an amplitude-weighted average of time. For visualization of ChIP-Seq coverage, .bigwig files were used and summarized into 10 bp buckets.

## 5. Conclussions

In summary, this analysis focuses on temporal enhancer and gene expression activation patterns that lead to novel observations. The highest NR3C1 recruitment is not directly associated with the highest dynamic in gene expression but rather precedes it. Correspondingly the highest dynamic in gene expression preceded the highest recruitment of EP300, which was unexpected considering the role of EP300 in enhancers activity. Furthermore, while gene expression dynamics declined, EP300 binding to chromatin stayed at high levels. In this orchestrated cascade of events, we observed an increase of H3K27 acetylation in NR3C1 responsive enhancers before maximum EP300 recruitment was reached and followed by increased H3K4me1 priming. These observations altogether show complicated temporal orchestration of various transcription mechanisms. The observations also point to possible positive feedback loops that might be involved in transcription regulation in case of response to repeated molecular stimuli. In such loops, initial NR3C1 binding would induce gene expression and chromatin remodeling as two distinct phenomena. The chromatin remodeling would then serve as a priming factor and lead to distinct responses to consecutive stimuli. This phenomenon is not well investigated. A recent study showed that GC exposure during neurogenesis leads to enhanced transcriptional responses to subsequent GC challenges [41]. Chromatin priming by NR3C1 could explain the role GC priming plays in regulating neuronal and glial responses to stress [42]. These priming loops could be involved in more complex processes, such as cell differentiation in which GR is widely known to be involved [43].

## Figures and Tables

**Figure 1 ijms-22-04258-f001:**
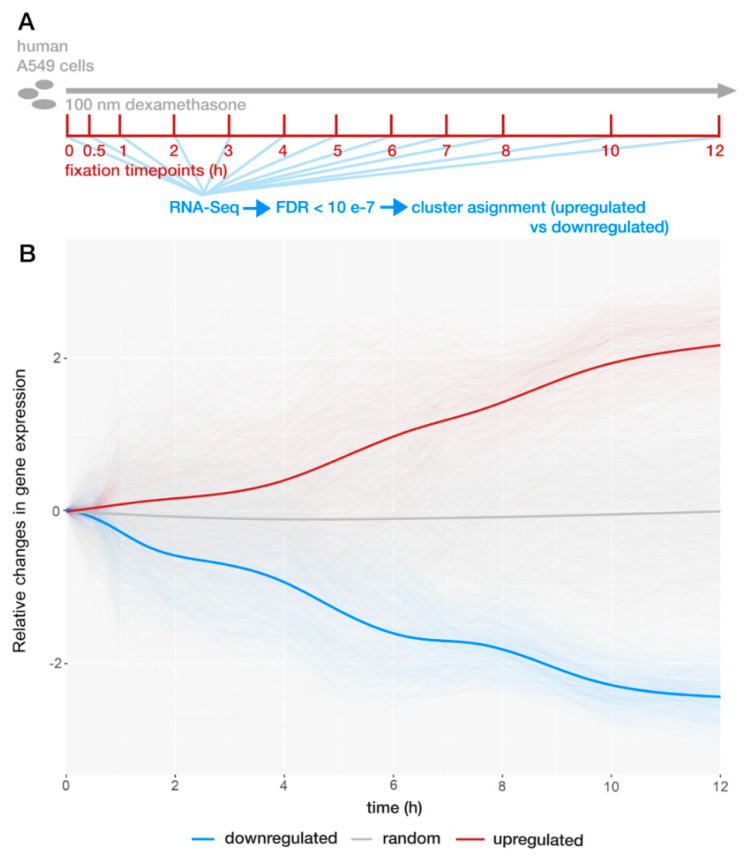
Profile of glucocorticoid receptor (GR)-induced gene expression changes. (**A**) Schematic representation of the experiment that was performed to generate the data analyzed in this publication. Data were downloaded from the ENCODE project database [6,21]. A549 epithelial-like human cell line was treated with 100 nM dexamethasone, and cells were fixed at distinct time points between 30 min and 12 h. RNA-Seq was performed at each time point. Raw RNA-Seq data were normalized and filtered for false discovery rate (FDR) < 0.0000001. (**B**) Abundance levels of transcript for each gene were scaled with z-score transformation and leveled to 0 at 0 min time point. Blue cluster—downregulated genes, red cluster—upregulated genes, gray cluster—not regulated genes. Appendix A contains a full list of regulated genes together with the *p*-value, FDR and cluster information. Appendix A lists the same information for selected random genes. Full raw data from the experiments are available from both ENCODE and GEO databases (see Materials and Methods).

**Figure 2 ijms-22-04258-f002:**
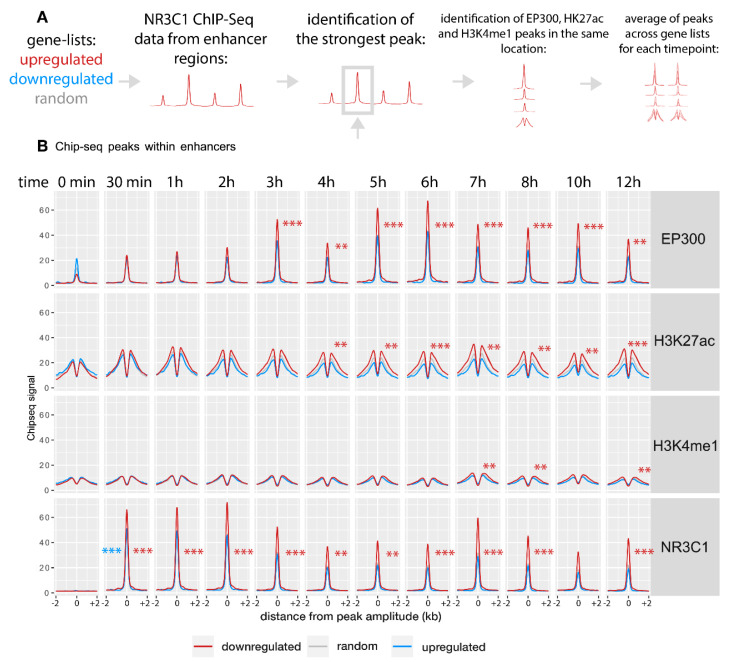
Time-course of average shapes of enhancer peaks associated with NR3C1. (**A**) Schematic representation of data analysis process. For each of the selected genes, the enhancer region was scanned for the strongest NR3C1 peak (+/− 100 kb with the exclusion of +/− 2 kb). EP300, H3K27ac, and H3K4me1 peaks were extracted from the same locations and averaged across genes from each cluster. (**B**) Each peak was centered according to the amplitude of NR3C1, and +/− 2 kb surrounding this position are plotted for each time point (average signal across genes). Two-way ANOVA time x regulation cluster: EP300: F = 22.6, *p* = 2 × 10^−6^; H3K27ac: F = 17.35, *p* = 3.2 × 10^−5^; H3K4me1: F = 14.96, *p* = 1.1 × 10^−4^; NR3C1: F = 6.43, *p* = 0.011 (also see Appendix A, Statistics: Appendix A). Stars represent the *p*-values of post hoc *t*-tests that compared amplitudes of peaks divided by the amplitude of random peaks in each cluster (upregulated: red stars, downregulated: blue stars) at each time point compared to the same value at 0 min. ** *p* < 0.01, *** *p* < 0.001.

**Figure 3 ijms-22-04258-f003:**
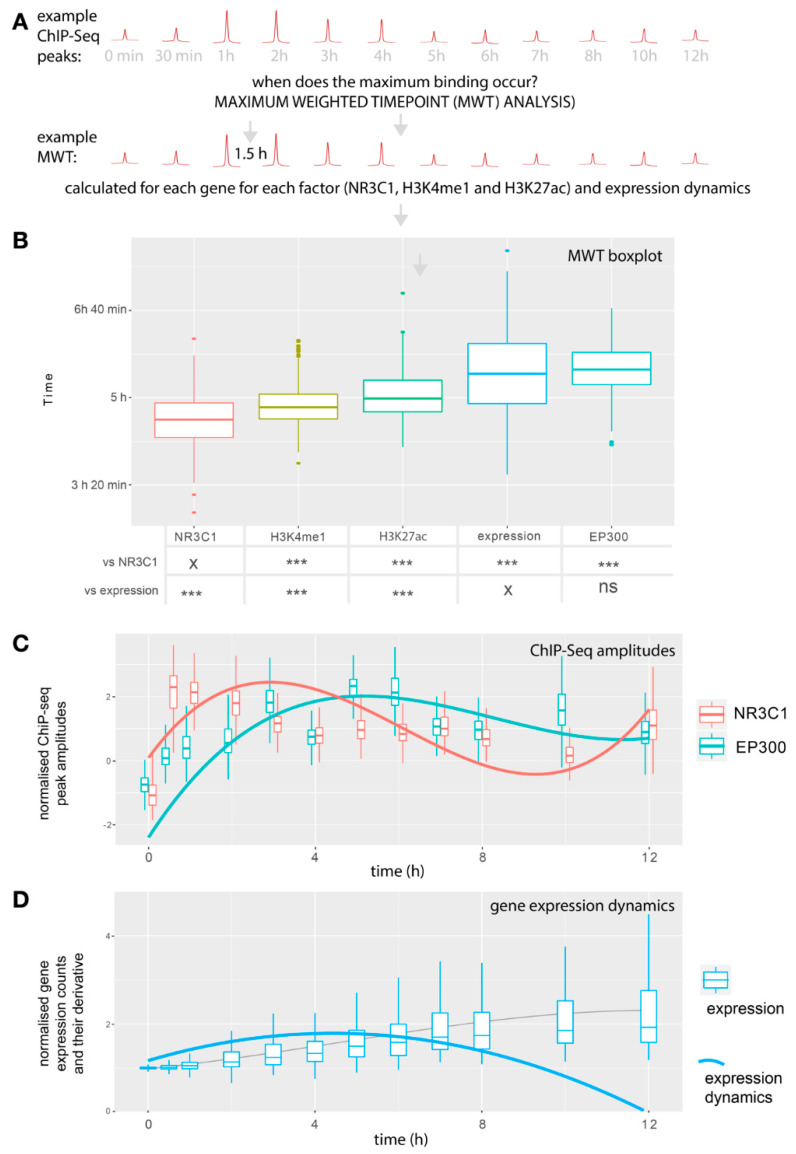
Analysis of gene expression dynamics and enhancer remodeling in upregulated genes with NR3C1 binding to enhancer sites. (**A**) Schematic illustration of weighted maximum time point (MWT) analysis. MWTs were extracted from analyzed ChIP-Seq regulatory factors and gene expression dynamics. For each factor, an average time-weighted by peak amplitude was calculated. For example, an MWT of 1.5 h means that for this factor, the maximum amplitude was observed both as 1 h and 2 h. For expression, MWTs were calculated based on the difference in transcript counts to illustrate expression dynamics. The higher the MWT, the later maximum binding/dynamics occur. (**B**) Boxplots of MWTs for all upregulated genes with identified enhancer peaks for each of the regulatory factors measured. For comparison of weighted time points, one-way ANOVA (F = 178.5, *p* = 2 × 10^−16^) with post hoc pairwise tests was used. Gene expression and NR3C1 weighted time points were tested against all other weighted time points with Bonferroni correction (expression vs. EP300:0.303, all other test *p*-values < 1 × 10^−4^, (See Appendix A for details). (**C**) Boxplot of z-scored amplitudes of ChIP-Seq peaks for NR3C1 and EP300 during the time course with fitted 3rd-degree polynomial lines. (**D**) Boxplot of gene expression counts represented as log2(count/(mean count at 0 min)). The fitted line does not represent changes in counts linearly as it is fitted to show expression dynamics. The scales of the boxplots and the fitted line are not directly comparable. Fitted line: A 3rd-degree polynomial fit to the derivative of gene expression scaled by a factor of 100 to equalize the scales). ** *p* < 0.01, *** *p* < 0.001.

**Figure 4 ijms-22-04258-f004:**
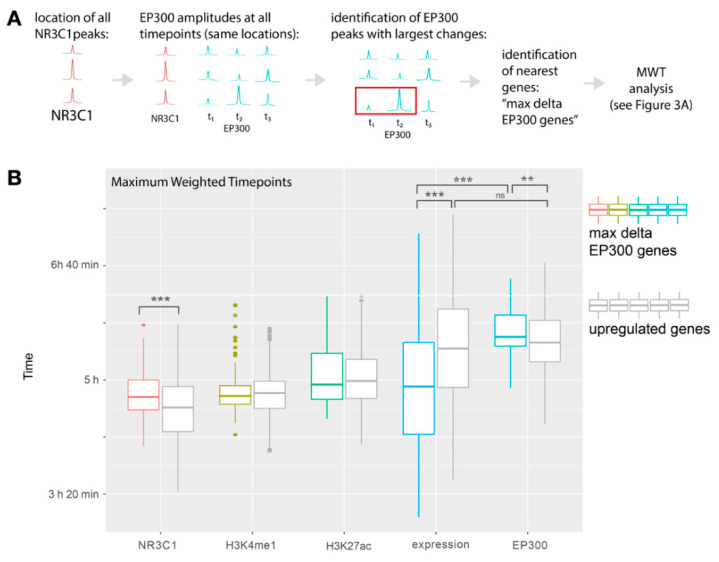
Analysis of gene expression dynamics and enhancer remodeling in upregulated genes with the highest changes in EP300 binding to enhancer sites. (**A**) Overview of the analysis steps. First, NR3C1 peaks were located. Next, EP300 peak amplitudes were collected. A list of 100 genes closest to the peaks with maximum EP300 changes was identified, and the upregulated genes from this list (*n* = 85) were subjected to MWT analysis (see Figure 3A). (**B**) Boxplots of MWTs for genes with largest EP300-binding changes (colored bars) compared with the data for upregulated genes (gray bars, same data as Figure 3B, gray bars). To compare weighted time points, two-way ANOVA with post hoc pairwise tests was used (See Appendix A and S9 for details). ** *p* < 0.01, *** *p* < 0.001 (*p* < 0.05 is not marked for clarity).

## Data Availability

Data used in this study are available via the ENCODE project.

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
