# Peer review of "Role of Non-Coding Regulatory Elements in the Control of GR-Dependent Gene Expression"

_ijms, 2021, doi:10.3390/ijms22084258_

Round 1
Reviewer 1 Report
In the paper entitled Role of non-coding regulatory elements in the control of GR dependent gene expression the authors investigated the spatial and temporal relation between GR binding to enhancer regions, chromatin remodelling and changes in gene expression of the genes upregulated after GR binding. The authors discovered that chromatin remodelling at the enhancer regions coincides with the highest changes in the target genes expression. To investigate the interplay between GR binding enhancer remodelling and gene upregulation, they analysed the ENCODE CHIP-seq and RNA-seq data from lung adenocarcinoma A549 cells. Based on their mathematical calculations the authors conclude, that GR binding to DNA and p300 binding to enhancer regions of the same genes don not coincide and that enhancer remodelling occurs app. 1h after GR binding and overlaps with the upregulation genes. It is an interesting paper that could be improved to have more clear narrative for non-bioinformatician. In the analysis of bioinformatics results, several approximations were taken into account which may impact the relevance of the presented results. It would contribute to the quality of the paper if the authors could provide some examples of their proposed GR-enhancer remodelling model. GR gene regulation has been extensively studied. Are their previously published results that align with their model? It would be beneficial and help the reader to understand their model if the authors would draw their model. I would suggests the paper for revision.
Figure 1. In the first Figure the authors show how the expression of the selected target genes changes with time after treatment with dexamethasone. Can the authors add more details to figure legend, such as, which cells, what kind of treatment of cells, cite the table where all the genes incorporated in the figure are listed? How do results correlated with previously published gene lists?
Figure 2. To determine what kind of correlation exists between gene upregulation and GR enhancer binding, the authors decided to examine enhancer regions with GRE sites, where GR was bound 1h after treatment. GR binding site +/- 100 kb and 2 kb (for proximal regions) were investigated. Figure S1.2. This figure should be better described. How have they decided for this time point? Would there be a different observation if other time points where chosen as well? Figure S2.1 is showing the number of gene in correlation with GR binding location (relative to TSS). Figure S2.1 legend should be better described. Their results show that most of the GR binding sites reside in the proximal promoter region. Their results in Figure S2.1 demonstrate that 1h after treatment with dexamethasone, most of the gene that are up-regulated or down regulated are due to GR binding to proximal promoter sites. In Figure 2 they further analysed GR binding sites in the enhancer regions. They examined if GR peak overlaps with ChIP-seq peak of P300, H3K27ac and H3K4me1. Based on their results, the dynamic of GR and other factors is not the same. Whereas GR reaches peaks from 1-2 h after treatment, binding of p300 reaches peak app. 6 h after treatment suggesting that chromatin remodelling occurs several hours after GR binding. What is represented in the graph? The authors should better describe the Figure 2.
Figure 3. In the Figure 3, the authors compared weighted time points of gene upregulation after addition of dexamethasone to cells and weighted time points of GR DNA binding. Based on their results, there is 1 h time lapse between GR binding and a possible consequent up-regulation of genes. The results in panel A should be presented clearer.
Figure 4. In the Figure 4, expression of genes was compared with p300 enhancer binding. This Figure also required more clear representation of the results.
Minor comments:
- Line 54 million base pairs?
- Line 131 dexamethasone treatments is missing; which time point was chosen at the first step
- Line 245: dexamethasone treatment
- All the references are in different font than the main text.
Author Response
The authors would like to thank the reviewers for their comments. We aimed to incorporate all the advice given by the reviewers. Changes in the manuscript are highlighted in yellow. Below is our point-by-point response.
Reviewer 1
In the paper entitled Role of non-coding regulatory elements in the control of GR dependent gene expression the authors investigated the spatial and temporal relation between GR binding to enhancer regions, chromatin remodelling and changes in gene expression of the genes upregulated after GR binding. The authors discovered that chromatin remodelling at the enhancer regions coincides with the highest changes in the target genes expression. To investigate the interplay between GR binding enhancer remodelling and gene upregulation, they analysed the ENCODE CHIP-seq and RNA-seq data from lung adenocarcinoma A549 cells. Based on their mathematical calculations the authors conclude, that GR binding to DNA and p300 binding to enhancer regions of the same genes don’t not coincide and that enhancer remodelling occurs app. 1h after GR binding and overlaps with the upregulation genes. It is an interesting paper that could be improved to have more clear narrative for non-bioinformatician.
To make the study more accessible to readers we have now included clarification statements throughout the manuscript (see responses below) and added exploratory panels to Figures 1, 2 and 3
In the analysis of bioinformatics results, several approximations were taken into account which may impact the relevance of the presented results. It would contribute to the quality of the paper if the authors could provide some examples of their proposed GR-enhancer remodelling model. GR gene regulation has been extensively studied. Are their previously published results that align with their model? It would be beneficial and help the reader to understand their model if the authors would draw their model. I would suggests the paper for revision.
Indeed, GR regulation has been extensively studied. However, our study focuses on the relation between: GR binding, chromatin remodelling and gene expression, with particular emphasis on the timing of maximum EP300 binding which is a novel area and, to the best of our knowledge, all the available literature related to this phenomenon has been cited. We have now extended the introduction and discussion by adding the following statements to better convey this information:
in the introduction:
GR effects on gene expression have been thoroughly investigated in a number of studies and extensively reviewed (Ramamoorthy & Cidlowski, 2016; Ratman et al., 2013; Sacta et al., 2016; Weikum et al., 2017). However it is not known how enhancer remodelling induced by NR3C1 connects to gene expression dynamics.
in the discussion:
In such loops, initial NR3C1 binding would induce gene expression and chromatin remodelling as two distinct phenomena. The chromatin remodelling would then serve as a priming factor and lead to distinct responses to subsequent stimuli. This phenomenon is not well investigated. A recent study showed that GC exposure during neurogenesis leads to enhanced transcriptional responses to subsequent GC challenge (Provençal et al., 2020).
Figure 1. In the first Figure the authors show how the expression of the selected target genes changes with time after treatment with dexamethasone. Can the authors add more details to figure legend, such as, which cells, what kind of treatment of cells, cite the table where all the genes incorporated in the figure are listed?
Figure 1 has now been updated, together with its legend and the reference to appropriate tables (Table S1.1 and Table S1.2 ) have been added.
How do results correlated with previously published gene lists?
We used EnrichR to identify overrepresentation of analysed genes in the previously published datasets. There were 346 published gene-sets with significant overlap (Adjusted P < 0.05). The list of overlapped publications include steroid dependent lists of genes as well as genes regulated in hormone dependent cancers:
- (PMID: 20035825) “Androgen receptor and androgen-dependent gene expression in lung”
- (PMID: 18662380) “Meta-analysis of gene expression profiles in breast cancer: toward a unified understanding of breast cancer subtyping and prognosis signatures
- (PMID: 16707422) ”Evolution of the androgen receptor pathway during progression of prostate cancer”
- (PMID 15897907) “Identification of molecular apocrine breast tumours by microarray analysis”
The above information is now included in the manuscript (results section). The list was also significantly overlapped with the original study of this dataset by Anthony M D'Ippolito et al. (PMID: 30031775).
Figure 2. To determine what kind of correlation exists between gene upregulation and GR enhancer binding, the authors decided to examine enhancer regions with GRE sites, where GR was bound 1h after treatment. GR binding site +/- 100 kb and 2 kb (for proximal regions) were investigated. Figure S1.2. This figure should be better described.
The description of Figure S1.2 is now updated and states:
Figure S1.2 Mean abundance (RPKM) histogram of analysed genes. In order to investigate whether there are any intrinsic differences in the average abundance of investigated transcripts for each transcript the normalised abundance (log2 (mean RPKM)) was calculated and averaged between all the time points. These values were then collected into a histogram and compared between each of the three clusters (up, down and random). x-axis: log2 of mean gene transcript count (RPMK), y-axis: fraction of transcripts in each bin (out of 1).
How have they decided for this time point? Would there be a different observation if other time points where chosen as well?
We selected a time point in which there is the most significant GR binding increase for both upregulated and downregulated genes (see Figure 2 and Figure S2.2). Peaks from subsequent time points overlap significantly. They differ mostly by amplitude. Thus, change of a time-point would have only marginal impact on final results.
This information has now been added into the manuscript:
in the methods section: This time point was chosen as it had the highest overall GR binding as compared with other timepoints.
Figure S2.1 is showing the number of gene in correlation with GR binding location (relative to TSS). Figure S2.1 legend should be better described.
We have now updated the figure S2.1 description and it states:
Figure S2.1 Histogram of GR strongest enhancer peak amplitude distance from transcription start site (TSS). For each of the differentially regulated genes a single NR3C1 binding site in an enhancer region of each gene was chosen for analysis. This single site was based on the strongest (highest amplitude) NR3C1 peak at the 60 min time point. Enhancer regions were determined as located between -100 to -2 kb and +2 to +100 kb from TSS (the core promoter region was excluded). y-axis represents distance from TSS in bp; x-axis - number of peaks in each 4 kb bin. Peaks from each of the clusters: up, down and random are summarised separately.
Their results show that most of the GR binding sites reside in the proximal promoter region. Their results in Figure S2.1 demonstrate that 1h after treatment with dexamethasone, most of the gene that are up-regulated or down regulated are due to GR binding to proximal promoter sites.
The above statements are only partially correct, we apologise for not clarifying the picture enough. Many of the GR binding sites were indeed located within promoters (the average GR signal from the promoter region is visualised in figures S2.4 and S2.5). What can be seen in Figure S2.1 is only what we consider to be enhancer peaks: for this analysis the regions -2 to +2 kb surrounding each TSS were excluded. Indeed many of the peaks tend to occur in the vicinity of core promoters.
We have now added the following statement to clarify the observations presented in Figure S2.1:
The distribution of identified enhancer peaks showed that in the case of up-regulated genes the highest peaks tend to occur closer to the promoter region than for the other gene clusters (Figure S2.1). Still, these detected peaks are well outside the core promoters, often defined as the region of a few hundred base pairs surrounding the transcription start site (TSS) (Liu & States, 2002).
In Figure 2 they further analysed GR binding sites in the enhancer regions. They examined if GR peak overlaps with ChIP-seq peak of P300, H3K27ac and H3K4me1. Based on their results, the dynamic of GR and other factors is not the same. Whereas GR reaches peaks from 1-2 h after treatment, binding of p300 reaches peak app. 6 h after treatment suggesting that chromatin remodelling occurs several hours after GR binding. What is represented in the graph? The authors should better describe the Figure 2.
Figure 2 now has an additional explanatory panel ‘A’ and a reviewed legend.
Figure 3. In the Figure 3, the authors compared weighted time points of gene upregulation after addition of dexamethasone to cells and weighted time points of GR DNA binding. Based on their results, there is 1 h time lapse between GR binding and a possible consequent up-regulation of genes. The results in panel A should be presented clearer.
Figure 3 now has an additional explanatory panel ‘A’ and a reviewed legend.
Figure 4. In the Figure 4, expression of genes was compared with p300 enhancer binding. This Figure also required more clear representation of the results.
Figure 4 now has an additional explanatory panel ‘A’ and a reviewed legend.
Minor comments:
- Line 54 million base pairs?
This line did not describe base pairs, it has now been clarified and states:
Once an enhancer is active it works to activate transcription of its targets via chromatin loops. Millions of such loops exist in the human genome (Jin et al., 2013)
Line 131 dexamethasone treatments is missing; which time point was chosen at the first step
This has now been corrected, the time point information is also redacted.
- Line 245: dexamethasone treatment
This is now corrected.
- All the references are in different font than the main text
This change in formatting has been introduced by the editor, in the text we have submitted the references are in the same format.
Reviewer 2 Report
This manuscript uses available genomics data to assess temporal aspects of glucocorticoid receptor dependent transcriptional activation, assessing the order to GR binding, coactivator binding, histone modification and transcription. Via this descriptives over time, they conclude that overall, p300 recruitment and histone modification occur later than would be expected if they where needed for transcriptional activation. This argues for a reinterpretation of our concept of simple coupling of all factor, but rather a primary transcriptional response, then followed later by a chromatin modification phase (that is: an activational and a priming effect).
The approach is nice, and the analyses seem solid. I do have some questions and comments.
Figure 2 really nicely makes a general point as in 'bulk' (many genes) events at GR loci. This figure suggests the order of GR - EP300, H3K27ac, H3K3me. The mathematical fit in figure 3 suggests another order (as assessed by a delta change over time value?): GR - histones - mRNA/EP300. And then the final analysis states that for the top-Chip-genes, only EP300 peaks late.
I find this confusing, also because in figure 1 ∆mRNA/∆time seems to be highest before 4 hours (but these are more genes). In itself it is not strange that different selection criteria lead to slightly different outcomes, but in particular the relative discrepancy of fig 3 confuses me.
I also do not understand why the fitted curve consistently is above the median value in figure 3C.
The analysis assumes a monophasic process, but there is a small blip for EP300 at 4 hours. It this something real? Does it point to two populations with different kinetics? Do the joint individual traces (like in fig 1), or more formal analyses tell us something about that.
I would be curious to know whether differently behaving peaks have any particular cis-element code? In other words: is it possible to do a motif analysis? Of course this can only be done when there are clear categories of loci/genes, as defined by temporal differences.
minor: I could not follow figure S2.1 - how does this relate to localisation of peaks?
Groups of genes are pooled, which makes statement like 'there is no point in which all TFs are acting in one machinery' rather vulnerable. This is on average.
Please not in the same sentence that histone modifications are not transcription factors - you are looking at one TF, one coactivator and two histone modifications
I hope the authors can help de-confuse me a little. All in all I think it is rather elegant to address a clear question like the present on in existing data.
Author Response
The authors would like to thank the reviewers for their comments. We aimed to incorporate all the advice given by the reviewers. Changes in the manuscript are highlighted in yellow. Below is our point-by-point response.
Reviewer 2
This manuscript uses available genomics data to assess temporal aspects of glucocorticoid receptor dependent transcriptional activation, assessing the order to GR binding, coactivator binding, histone modification and transcription. Via this descriptives over time, they conclude that overall, p300 recruitment and histone modification occur later than would be expected if they where needed for transcriptional activation. This argues for a reinterpretation of our concept of simple coupling of all factor, but rather a primary transcriptional response, then followed later by a chromatin modification phase (that is: an activational and a priming effect). The approach is nice, and the analyses seem solid. I do have some questions and comments.
Figure 2 really nicely makes a general point as in 'bulk' (many genes) events at GR loci. This figure suggests the order of GR - EP300, H3K27ac, H3K3me. The mathematical fit in figure 3 suggests another order (as assessed by a delta change over time value?): GR - histones - mRNA/EP300. And then the final analysis states that for the top-Chip-genes, only EP300 peaks late.
The question we have used to analyse the dynamics was the following: At which time point does the MAXIMUM binding of each factor occur? (see Figure 3A for clarification). Although Figure 2 (and Figure S2.2) indeed show and early rise of EP300 binding the highest binding occurs, on average, much later and this is the value assessed in Figure 3.
I find this confusing, also because in figure 1 ∆mRNA/∆time seems to be highest before 4 hours (but these are more genes). In itself it is not strange that different selection criteria lead to slightly different outcomes, but in particular the relative discrepancy of fig 3 confuses me.
We apologise for the confusion. In Figure 1, if one looks at the thicker red line of the up-regulated genes the slope (and thus derivative) is the steepest between 4-6 h. In Figure 3 indeed some (significant outliers) genes were excluded from the analysis, but the highest dynamics (largest derivative) still fall between 4-6 h. We have updated all the Figure descriptions and added explanatory panels (“A” parts) to all the figures to make the manuscript less confusing.
I also do not understand why the fitted curve consistently is above the median value in figure 3C.
(Now Figure 3D). This is because the boxplot shows normalised gene expression and the line represents the derivative and is additionally on a different scale. This information has now been added to the Figure legend (Part 3D) that states:
Boxplot of gene expression counts represented as log2(count/[mean count at 0 min]). The fitted line does not represent changes in counts linearly as it is fitted to show expression dynamics. The scales of the boxplots and the fitted line are not directly comparable. Fitted line: a 3rd-degree polynomial fit to the derivative of gene expression scaled by a factor of 100 to equalize the scales). ** p < 0.01, *** p < 0.001
The analysis assumes a monophasic process, but there is a small blip for EP300 at 4 hours. It this something real? Does it point to two populations with different kinetics? Do the joint individual traces (like in fig 1), or more formal analyses tell us something about that.
Unfortunately there is no way of knowing whether this ‘blip’ is a biological phenomenon or a random event due to inherent noise of the data. Figure S2.3 shows a heatmap of individual EP300 peaks (each line represents one peak over time) and there are no indications of two populations of peaks. This information has now been added to the results section:
EP300 binding near up-regulated showed a notable decrease in signal at the 4 h timepoint. This could be indicative of overlapping of distinct patterns of EP300 binding but individual time-courses did not confirm this observation (Figure S2.3).
I would be curious to know whether differently behaving peaks have any particular cis-element code? In other words: is it possible to do a motif analysis? Of course this can only be done when there are clear categories of loci/genes, as defined by temporal differences.
This is indeed a very interesting question, but as we did not observe distinct “peak populations” in our analyses we could not address it. In our exploratory analyses we did aim to identify distinctly regulated clusters of genes, namely up-regulated early, up-regulated later e.t.c. but there were no major differences in temporal or dynamics between these clusters (nor in the binding amplitudes).
minor: I could not follow figure S2.1 - how does this relate to localisation of peaks?
This figure now has an updated legend and is better described in the results section:
in the results section: The distribution of identified enhancer peaks showed that in the case of up-regulated genes the highest peaks tend to occur closer to the promoter region than for the other gene clusters (Figure S2.1). Still, these detected peaks are well outside the core promoters, often defined as the region of a few hundred base pairs surrounding the transcription start site (TSS) (Liu & States, 2002).
Figure legend:
Figure S2.1 Histogram of GR strongest enhancer peak amplitude distance from transcription start site (TSS). For each of the differentially regulated genes a single NR3C1 binding site in an enhancer region of each gene was chosen for analysis. This single site was based on the strongest (highest amplitude) NR3C1 peak at the 60 min time point. Enhancer regions were determined as located between -100 to -2 kb and +2 to +100 kb from TSS (the core promoter region was excluded). y-axis represents distance from TSS in bp; x-axis - number of peaks in each 4 kb bin. Peaks from each of the clusters: up, down and random are summarised separately.
Groups of genes are pooled, which makes statement like 'there is no point in which all TFs are acting in one machinery' rather vulnerable. This is on average.
We have now included the following statement in the discussion:
Although strongly supported by the date, the above conclusions have limitations as they describe average dynamics of gene regulation and not individual time courses.
Please not in the same sentence that histone modifications are not transcription factors - you are looking at one TF, one coactivator and two histone modifications
This is a clear mistake and has now been corrected throughout. Whenever all four analysed ChIP-Seq tracks are mentioned the term we use is : regulatory factors.
I hope the authors can help de-confuse me a little. All in all I think it is rather elegant to address a clear question like the present on in existing data.
Multiple adjustments to the manuscript have been made in order to make the presented analyses clearer.
Reviewer 3 Report
The manuscript describes the temporal interplay between GR binding, enhancer remodeling, and gene expression. By investigating data from the ENCODE project, ChIP-Seq and RNA-Seq data were analyzed for the timing of interaction between protein_gene expression-transcription factor activation and histone modifications (H3K27ac and H3K4me1) following dexamethasone activation. Authors focused on a single ChIP-seq peak in the enhancer region, with the strongest NR3C1 amplitude by identifying peaks in terms of EP300 and NR3C1 binding and two histone modifications H3K27ac and H3K4me1. Authors well describe that the transcriptional changes follow GR binding and that the largest enhancer remodeling coincides in time with the highest gene expression changes, and results showed that the highest NR3C1 recruitment is not directly associated with the highest dynamic in gene expression but rather precedes it.
This is a well designed and precise study based on the DNA-GR interaction analysis by RNA seq related to ChIP seq data, which would help the understanding of the GR mechanisms on the DNA expression regulation.
Minor points:
lines 307-308: different text format
Author Response
The authors would like to thank the reviewers for their comments. We aimed to incorporate all the advice given by the reviewers. Changes in the manuscript are highlighted in yellow. Below is our point-by-point response.
lines 307-308: different text format
This change in formatting has been introduced by the editor, in the text we have submitted these lines are formatted the same way.